# Job Satisfaction Among Midwives in High-Intervention Birthing Rooms: A Qualitative Phenomenological Study

**DOI:** 10.3390/healthcare13111318

**Published:** 2025-06-02

**Authors:** Marta Pérez-Castejón, Laura Martínez-Alarcón, Alonso Molina-Rodríguez, Ismael Jiménez-Ruiz

**Affiliations:** 1Doctoral Programme in Health Sciences, International School of Doctoral Studies, University of Murcia, 30120 El Palmar, Spain; marta.perez@um.es; 2Murcian Health Service, 30100 Murcia, Spain; lma5@um.es (L.M.-A.); alonso.molina@um.es (A.M.-R.); 3Department of Nursing, Faculty of Nursing, University of Murcia, 30120 El Palmar, Spain; 4Advanced Nursing Care (Cuidados Avanzados en Enfermería), Murcian Institute of Biomedical Research, Virgen de la Arrixaca, 30120 El Palmar, Spain

**Keywords:** midwifery, midwives, job satisfaction, midwifery autonomy, intrapartum, collective identity

## Abstract

**Objectives:** To identify the factors influencing the job satisfaction of midwives working in birthing rooms with a medium to high level of obstetric intervention. **Methods:** A qualitative phenomenological–hermeneutic design based on Heideggerian philosophy was implemented. A study involving 25 participants, midwives, and resident nurses (RINs) was conducted. Residents were included to enrich the analysis with their critical perspectives with regard to interventional procedures and exposure to acts of obstetric violence. Convenience sampling was used. Data were collected from four focus groups in three hospitals in the region and one in-depth interview between 30 September 2022 and 23 June 2023. The testimonies were recorded and transcribed verbatim, and data analysis was carried out using an inductive–deductive approach. The triangulation of data and researchers was used to minimise potential bias. **Results:** The participants highlighted the following key dimensions or areas that contribute to midwives’ job satisfaction in the labour and birthing room: maternal satisfaction, professional competencies, multidisciplinary team, working conditions, and interventions during childbirth. **Conclusions:** These findings may inform healthcare management strategies to reduce burnout and improve working conditions in maternity care settings.

## 1. Introduction

Job satisfaction is a key factor for the well-being of healthcare professionals and is closely linked to the quality of care they provide [1,2]. In the healthcare sector, work environments can be especially demanding, often exposing professionals to significant physical and emotional risks [3,4]. Supporting the job satisfaction of these professionals not only benefits their health but also enhances outcomes for patients, families, and healthcare systems [4].

Midwives’ job satisfaction is of growing concern, as it affects the safety of the care provided to women and newborns during pregnancy, childbirth, and the postpartum period [5,6]. In particular, in maternity units, professionals must be competent and use the best available scientific information to care for the mother during labour and to monitor the condition of the foetus in utero and the newborn after birth [7].

The burden of care and the physical demands of the job in midwifery have increased with the growing medicalisation of perinatal care [8]. This also leads to power imbalances between the professionals involved in childbirth and a lack of professional autonomy and recognition for midwives, affecting their job satisfaction [9]. It also promotes practices that are far from safe for expectant mothers and their newborns [10,11]. Therefore, a midwifery model of care for low-risk pregnancies and births offers advantages over this medicalised model, including a positive birth experience for women, peace of mind, enhanced well-being, and increased job satisfaction, as well as reducing the occurrence of obstetric intervention [12].

Midwives who suffer from low job satisfaction may experience physical and mental health issues, as well as a decline in performance, physical and mental exhaustion, despair, and frustration [13]. This can lead to psychiatric conditions, such as depression or anxiety [14], and even to burnout and career withdrawal, which may compromise the safety and quality of the care delivered to society [15].

Job satisfaction among maternity professionals has been the subject of much research worldwide. It is well-established that the working environment, length of service, long working hours, lack of autonomy, challenges around work–life balance and professional recognition, poor pay, and a lack of support from management and colleagues are all factors that impact well-being at work [16]. Lack of time and space for multidisciplinary clinical sessions, or “huddles”, is also known to be an important factor [17].

Although numerous studies have explored factors affecting job satisfaction in midwifery, most have taken a quantitative approach focused on measuring predefined variables, such as salary, workload, and work–life balance [16]. These studies, while valuable, may overlook the deeper subjective experiences of midwives working in highly medicalised birth settings. Moreover, the measurement tools employed in these studies to assess midwives’ job satisfaction often fail to consider domains related to the medicalisation of childbirth, exposure to obstetric violence, and role conflict among professionals involved in intrapartum care [9]. This could explain why midwives working in birthing centres tend to report higher levels of job satisfaction compared with those practising in hospital settings [18]. Furthermore, this omission may shed light on the apparent contradiction whereby studies evaluating job satisfaction among hospital-based midwives report adequate levels of satisfaction, while research focusing on professional attrition and burnout among the same population reveals alarmingly high rates [16,19].

Given the complexity of this phenomenon, a qualitative hermeneutic approach allows for an in-depth exploration of how midwives understand and give meaning to their work experiences. This interpretive framework is particularly suited to uncovering the nuances and contextual factors that shape job satisfaction in environments where technical interventions are frequent and decision making may be constrained. Thus, this study aims to explore the lived experiences of midwives working in labour wards with a medium to high level of obstetric intervention in order to identify the factors that influence their job satisfaction.

## 2. Materials and Methods

### 2.1. Study Design

A qualitative study with a phenomenological design based on a hermeneutic approach was conducted to explore and understand the factors affecting job satisfaction among midwives working in delivery rooms. A hermeneutic phenomenological perspective, influenced by Heideggerian philosophy, was chosen, because it facilitates a deeper understanding of how individuals make sense of their experiences within a specific context [20]. To this end, discourses were analysed and interpreted with the aim of understanding each lived experience as the foundation of this research [21]. The document was prepared following the consolidated criteria for reporting qualitative research (COREQ) [22].

### 2.2. Participants

The participants were midwives working in the maternity units of different hospitals in the Region of Murcia (Spain). A midwife from a public hospital in Argentina and second-year midwifery students (resident intern nurses (RINs)) were also involved. These midwifery residents are in their final year of training; therefore, they do not require strict physical supervision, and their roles are similar to those of a qualified midwife. Their inclusion in this study was based on their integration into the work teams and their exposure to the same working environment, thereby providing relevant insights into job satisfaction during the transition to full professional autonomy. As for the inclusion of a midwife from Argentina, they were included through an in-depth interview with the intention of broadening the analytical lens and identifying any context-dependent variations in professional satisfaction. This participant’s testimony was analysed separately and used illustratively, without influencing the main thematic saturation drawn from the focus groups conducted in Spain.

We used convenience sampling and gained access to the study population via the supervisors of each maternity unit and the authors’ personal networks [23]. Given the qualitative and exploratory nature of the study, convenience sampling was chosen as a suitable methodological strategy to identify and access participants who were available and able to provide rich and meaningful accounts of their professional experience. This type of sampling is widely accepted in phenomenological studies, where the primary goal is the depth and richness of analysis rather than statistical representativeness. Consequently, we set the eligibility criteria to ensure that the characteristics of the participants were closely matched to the aims of the study. Individuals who were on sick leave at the time of the study were excluded. The sample size was determined on the basis of information saturation, in accordance with the guidelines established by Malterud [24]. Data collection was halted once thematic saturation was reached—that is, when the inclusion of additional participants no longer yielded new categories or substantially enriched the understanding of the phenomenon under study.

Although the invitation to participate was extended to all midwives working in the maternity units of the hospitals included in the study, in the end, 14 midwives and 10 RINs chose to participate in the groups. We held three focus groups with midwives and a focus group discussion with final year RINs from the above hospitals. Finally, we conducted an in-depth interview with a midwife from an Argentinian public hospital to gain a broader perspective on the research question.

### 2.3. Data Collection Procedure

A semi-structured interview guide was developed based on theoretical and empirical literature on job satisfaction among healthcare professionals, midwifery models of care, and workplace dynamics in maternity services [6,10,11,25,26,27,28]. The guide included nine thematic areas (e.g., job satisfaction, workload, professional autonomy, multidisciplinary team, etc.), each accompanied by one or more core questions, as shown in Appendix A. In total, 17 main questions were formulated to initiate discussion. These were complemented by probing or follow-up questions as needed, allowing participants to elaborate on their experiences and perspectives in a flexible yet structured manner.

The principal researcher was chosen to guide the reflection in the focus groups because of her expertise in qualitative research and her ability to foster an open and trusting environment for discussion. Her role was essential in facilitating the in-depth exploration of participants’ experiences, ensuring that their narratives were accurately interpreted and contextualised. This approach aligns with the criterion of credibility. Throughout the focus groups, the interviewer’s involvement was limited to answering any questions that arose about the issues being discussed. This allowed the participants to reflect critically on their experiences in retrospect, providing an in-depth understanding of their experiences in terms of job satisfaction. At the end of the groups, the researcher led a reflective process with the participants, with a view to clarifying their accounts by paraphrasing and summarising their stories to ensure that they were properly understood.

The lead researcher arranged the venues for the focus groups to suit the participants. The midwife focus groups were held in the staff rooms of the maternity units, and the in-depth interview was conducted in the Faculty of Nursing at the University of Murcia (UM). Informal discussions with the participants were held beforehand in preparation for the group process. The duration of the focus groups (15–25 min) was determined by the availability of participants within the context of the birthing ward, where clinical duties and workflow shaped the timing of the sessions. The discussions were intensive and guided by a semi-structured script grounded in theory, allowing participants to express their experiences meaningfully and reflectively. This approach, together with complementary field notes and an in-depth interview, ensured phenomenological depth despite time constraints.

Data collection was discontinued upon reaching data saturation. Data were transcribed and analysed after each focus group was conducted. This allowed us to incorporate new thematic lines that had emerged in the subsequent focus group. This process continued until focus group number 4, in which no new emerging categories were found. At this point, the collection of new data did not change our coding manual.

### 2.4. Data Analysis

We used Atlas.ti 25 software for the extraction, organisation, and classification of data. The analysis followed a thematic approach, combining inductive and deductive strategies as proposed by Fereday and Muir-Cochrane [29]. This methodological integration allowed us to remain grounded in the participants’ narratives, while also drawing on existing literature to refine our understanding.

Two researchers independently read and re-read the interview transcripts to become deeply familiar with the content. Initial codes were generated inductively, allowing themes to emerge from the data. The coding process was conducted independently to ensure analytical openness and minimise individual bias. Subsequently, both coders met to compare their findings. Discrepancies were discussed in depth and resolved through consensus; when needed, a third researcher was consulted. All decisions were theoretically informed and supported by relevant literature to ensure credibility and rigour.

After initial coding, the researchers organised the codes into preliminary categories based on thematic similarity. These categories were then iteratively reviewed and refined through constant comparison across transcripts. Particular attention was paid to identifying patterns related to midwives’ job satisfaction and its influencing factors, as these themes repeatedly emerged in the data. This inductive process was enriched by a deductive comparison with concepts previously identified in the literature review, allowing for further refinement and redefinition of the themes and categories.

Finally, the team examined the interrelationships among categories to construct a coherent thematic structure.

### 2.5. Rigour and Analytical Quality

To ensure the rigour and trustworthiness of the study’s findings, various strategies were implemented to strengthen credibility, transferability, and confirmability.

Credibility was enhanced through investigator triangulation via a systematic and collaborative process, prolonged engagement with the data, and regular discussion sessions among the research team members. This approach helped ensure interpretative depth and minimised potential individual biases.

Regarding transferability, rich and contextualised descriptions were provided for both the participants and their professional environments, as well as the healthcare system in which they operate. Additionally, a critical assessment of the contextual applicability of the findings was conducted, taking into account the specific characteristics of the sample and the study setting. Although the sample size is acknowledged as a potential limitation, the research team emphasised the relevance of the results for similar contexts and populations. This reflective approach allowed for context-sensitive interpretations with the potential to be transferred beyond the immediate scope of the study, which focused on job satisfaction among midwives in delivery rooms in Spain.

Finally, confirmability was ensured through the maintenance of a systematic audit trail documenting analytical decisions made throughout the research process. Reflexive memos were also developed to make the researchers’ positionality explicit and critically examine how it might have influenced the interpretation of the data.

### 2.6. Ethical Considerations

The study was approved by the Ethics Committee of the University of Murcia (approval reference CI 4065), following the standard coding system of the institution. The research was conducted in full accordance with the ethical principles set forth in the Declaration of Helsinki, including respect for autonomy, informed consent, confidentiality, and the right to withdraw at any time.

## 3. Results

Four focus groups and an in-depth interview were conducted. Of the 25 participants, two were male, ten had permanent contracts, five had temporary contracts, and ten had RIN training contracts. The midwives ranged in age from 30 to 66 years, and the midwifery RINs from 24 to 38 years (Table 1).

Figure 1 shows the interrelationship between categories. These include categories and subcategories relating to the factors influencing midwives’ job satisfaction, a definition of this concept, and the signs and symptoms observed in staff when they are dissatisfied. These themes are analysed below and include maternal satisfaction, midwifery skills in the birthing room, multidisciplinary teams, the childbirth process, and working conditions. The job satisfaction of midwives in midwifery is influenced by different dimensions. The dimensions as a whole and the interrelationship between them shape job satisfaction. The job satisfaction of midwives in midwifery is directly dependent on their working conditions. These working conditions are strongly influenced by professional development and job autonomy in the performance of the function. Respect for the competencies of the profession by the multidisciplinary team and teamwork have a decisive influence on the job satisfaction of midwives. Similarly, women’s satisfaction with the delivery process and obstetric outcomes shape midwives’ satisfaction with their work in the delivery room.

### 3.1. Definition of “Job Satisfaction”

For midwives, job satisfaction comprises several components identified in the focus groups, including psychological well-being, team belonging, commitment to care and the mother–child bond, enjoyment of work, and recognition from colleagues and the wider team.


*“Well, for me, satisfaction comes from… well, the feeling that we are part of a team, both on a… I mean, getting feedback from my colleagues, from the gynaecologists and *vice versa*, from my superiors (…) knowing that I have that support…”*
(P9)

From this definition, we can extract the following different dimensions or key areas highlighted by the participants that contribute to the job satisfaction of midwives in the birthing room: interventionism during childbirth, maternal satisfaction, professional competencies, multidisciplinary team, and working conditions.

### 3.2. Childbirth Process

The focus groups revealed that, for some midwives, job satisfaction depends more on positive obstetric outcomes than on the type of birth.


*“The goal is a positive obstetric outcome, whether it is a natural birth, a ventouse birth, or a caesarean section”.*
(P15)

Participants made repeated reference to the use of interventions that reduce medicalisation, increase humanisation, and improve obstetric outcomes as being fundamental to improving midwife satisfaction in the birthing room. Examples include the use of “walking epidurals”, non-pharmacological pain relief methods that reduce maternal anxiety and boost well-being, and the adoption of the “one-to-one” approach to improve woman-centred care.


*“What we don’t have is one-to-one care (…) of course it would be better and our satisfaction would probably improve. And it should be a goal that we strive for, to strive for one-to-one… that way of working”.*
(P8)

Midwives’ groups agreed on the need to review the correct application of protocols on the maximum time limits for active labour leading to obstetric interventions in order to comply with WHO recommendations.


*“If they’d just let women take their time… but they don’t give them time either…”*
(P11)

Identifying and witnessing acts of obstetric violence against women in active labour takes a significant psychological toll on their own perceived job satisfaction.


*“And… you can also feel like an ‘accomplice’ when you work with someone who has an attitude… bordering on obstetric violence or actual obstetric violence… and you can’t do anything about it… or you feel you can’t do anything to stop it…”*
(P19)

### 3.3. Maternal Satisfaction

Job satisfaction for both midwives and RINs is closely tied to maternal satisfaction during childbirth. Participants described how these concepts overlapped, leading to the integrated idea of maternal and midwife satisfaction, defined as the influence of maternal satisfaction on midwives’ perceived job satisfaction.


*“If a woman is well cared for, with everything that… whatever she needs and is offered… for the midwife caring for her, it’s a great source of satisfaction. It means working in a more comfortable, warmer environment and being able to give her even better care…”*
(P7)

In this context, midwives and RINs identified evidence-based care practices that influence maternal satisfaction, including holistic perinatal care, respect for autonomy, informed decision making, reduced interventionism, and the humanisation of necessary procedures. These themes, drawn from verbatim quotes, are closely linked to both maternal and midwife satisfaction (Table 2).

### 3.4. Midwifery Skills in the Birthing Room

Midwives’ job satisfaction is strongly associated with the ability to exercise their legally recognised competences and professional autonomy. A work environment that undermines professional development or disregards midwives’ clinical decisions diminishes their satisfaction and fosters frustration.


*“Imagine a doctor showing up all of a sudden and you’ve told the woman ‘I’m not going to break your water unless XYZ happens’ and then in comes the doctor and goes ‘wham’ and breaks her water… well, that sort of thing… well… it gets on your nerves”.*
(P11)

Participants, particularly those working in regional hospitals or in contexts where midwives enjoy greater autonomy, emphasise the importance of having the respect of the multidisciplinary team. This respect is reflected in the ability to carry out their functions autonomously and to feel valued in their role.


*“Here we have a lot of autonomy and we get a lot of respect… way more than in other hospitals… I’ve worked in other hospitals and… The doctor looks on but doesn’t do anything… they only intervene when the midwife says there’s a complication…”*
(P7)

This perspective implies not only attending to the technical aspects of the job but also to the biopsychosocial needs of the woman. Humanising care in the birthing room is essential for these professionals, with the priority being to provide holistic care and support to meet all the woman’s needs during childbirth and the postnatal period.


*“In the whole person, not just in the technical things you have to do, this thing or that thing… but in being there for that woman, supporting her in everything she needs…”*
(P20)

### 3.5. Multidisciplinary Teams

Effective communication between team members and relationships based on trust, respect, cohesion, backup, professional recognition, and support are all key pillars in fostering an environment conducive to professional development and job satisfaction.


*“To feel comfortable (…) to feel that (…) you’re valued…”*
(P10)


*“Yeah, when you say something and your gynaecologist colleague says… ‘I think you’re right’. You know, that they’re not questioning you”.*
(P13)

RINs in the study emphasised the negative impact of a dysfunctional or non-collaborative multidisciplinary team that fails to recognise midwives’ skills. Ineffective communication fosters mistrust, interference, and inequality, leading to parallel working practices that diminish both productivity and job satisfaction.


*“At the end of the day, teamwork is key… because if you’re seeing the woman and then the gynaecologist is there too, seeing the same woman, and they don’t say anything to you or ask you anything… or… (…) so… what’s the point?”*
(P21)

Both midwives and RINs emphasise the need for regular multidisciplinary training and suggest incorporating joint clinical sessions or “huddles” to enhance team confidence and manage critical situations.


*“I think these kinds of sessions would be really useful… as a group, multidisciplinary… on protocols to… um… to review protocols or things that are being rolled out and not just find out when you go to work or see stuff in the WhatsApp group, but something more like that for everyone…”*
(P19)

### 3.6. Working Conditions

Factors influencing job satisfaction at work include shift patterns, long working hours, excessive administrative duties, and inadequate pay. Extended shifts, particularly those lasting 12 or 24 h, are perceived as physically and mentally draining due to the demanding nature of maternity care.


*“12 or 24 h cooped up in here is awful…”*
(P10)


*“We also have a lot of admin, don’t we…?”*
(P12)


*“Yes… the… the salary increase also has a lot to do with it…”*
(P7)

They highlighted the importance of rooms with natural light, more physical space, and more neutral décor. They also mention the need for multidisciplinary rest and relaxation areas that encourage interaction between team members.


*“I feel like I’m in a dungeon (…) You need light, you need to go out… Sometimes I go out for breakfast or lunch just to see the light…”*
(P10)


*“(…) a small room or a shared office with the gynaecologists… not for any specific reason, but because it would mean less separation between our two professions and we’d feel more like a team…”*
(P18)

## 4. Discussion

Our findings contribute to the existing literature and highlight the following important factors that determine midwives’ job satisfaction in the birthing room: the childbirth process, maternal satisfaction, midwifery skills in the birthing room, the multidisciplinary team, and working conditions. Midwives appear to experience greater satisfaction when women’s expectations of the childbirth process are met and when they feel cared for and respected. They reported that highly interventionist settings diminished their job satisfaction, as these environments also reduced the satisfaction of the women in their care. Exposure to acts of obstetric violence also diminished their job satisfaction. Moreover, we found a bidirectional relationship between midwives’ job satisfaction and women’s satisfaction during childbirth; a lack of recognition of professional competencies and occupational autonomy; a need to work within a multidisciplinary team grounded in respect, effective communication, and mutual trust; and the need for more homely environments within hospitals, alongside improved working conditions, such as shorter working hours and better remuneration.

In terms of the childbirth process, the midwives in our study reported lower job satisfaction when women perceived the care they received as medicalised, interventionist, and dehumanising. These findings are consistent with those reported by Westergren et al. [30], who conducted a cross-sectional study aimed at exploring the medicalisation of childbirth through the use of pain relief methods and its relationship with satisfaction with the childbirth process. The sample included a total of 129 women with a birth plan and 110 without one. The results indicated that women who underwent more obstetric interventions and received more pharmacological pain relief methods reported lower levels of satisfaction with the childbirth process. Similarly, Wangler et al. [31] conducted a cross-sectional study to investigate whether low-intervention birth room design was associated with higher levels of job satisfaction among midwives. The study included a total of 312 midwives and found that they reported greater job satisfaction in low-intervention environments with fewer staff members in the multidisciplinary team.

Exposure to acts of obstetric violence emerged as another key topic discussed during the focus groups and was predominantly raised by RINs. This exposure to acts of obstetric violence has also been examined by Mena-Tudela et al. [32] in a qualitative study where, in line with our findings, professionals stated that they were confronted with acts of violence against women on a daily basis, even during their training. In the same vein, the findings of the study conducted by Martínez-Galiano et al. [33] are also relevant. This study employed a descriptive qualitative design to assess midwives’ knowledge and practices that might be associated with obstetric violence, as perceived by the midwives themselves. The results revealed that many midwives engaged in obstetric violence due to pressure from medical colleagues, high workloads, or in some cases, simply witnessed such acts carried out by other professionals. In both situations, the midwives expressed a sense of complicity and reported bearing a “horrible emotional burden”. In line with these findings, Olza [34] also indicates that perpetrating or witnessing obstetric violence leads to feelings of demotivation, guilt, helplessness, and deep sadness or anger, including during training.

In terms of maternal and midwife satisfaction, we identified limited evidence directly supporting this relationship in hospital care. Our results align with those obtained in the study by Mashayekh-Amiri et al. [35], who showed how midwives feel useful and professional when working with pregnant women and establish a strong, supportive relationship. In accordance with these findings, Sadiku et al. [36] examined the quantitative relationship between group care and overall maternal satisfaction compared with standard individual care through a systematic review. Although in this case satisfaction was examined in the prenatal period, one of the findings was the “midwife effect”, being associated with our “maternal and midwife satisfaction”. This term indicates that when women receive the care they desire and require from midwives, they experience satisfaction with the perinatal processes, which in turn fosters a sense of professional fulfilment in the attending midwife.

In accordance with the findings from our results regarding the multidisciplinary team and working environment, the systematic review conducted by Pérez-Castejón et al. [16] showed that a lack of professional recognition and role conflicts limited midwives’ job satisfaction. Similar findings were reported by Andina-Díaz et al. [9] and Mashayekh-Amiri et al. [35]. Likewise, the study conducted by Mharapara et al. [10] found that having the autonomy to make independent decisions, as well as midwife empowerment, had a positive effect on job satisfaction and professional development. These outcomes were reported to be more pronounced among midwives working in birthing centres compared with those in hospitals. Similarly, the study by Vermeulen et al. [37] revealed that, although participants rated their professional autonomy as high, midwives expressed a desire for greater recognition and respect from society and the wider multidisciplinary team.

In the case of working conditions, some authors, such as Rodríguez-García et al. [38], reported findings consistent with ours, as they showed that when working conditions are unfavourable, the safety of the care provided is reduced and there is greater intent among midwives to leave their job and even the profession. In the same vein, in the study by Aikins et al. [39], midwives were dissatisfied with their salaries, while Hansson et al. [40] identified a lack of organisational resources as a modifiable factor affecting midwives’ satisfaction. Additionally, Şahan et al. [6] conducted a cross-sectional study to determine the level of job satisfaction among midwives and the factors influencing it and noted that midwives perceived long working hours as exhausting and harmful to physical and mental well-being, making it difficult to empathise and humanise the care work. Long working hours can also be a risk factor for the onset of burnout. Finally, authors such as Wangler et al. [31] concur that midwives working in birthing centres with more homely rooms reported higher levels of job satisfaction than midwives working in conventional hospitals. This can be attributed to the fact that homely environments are more conducive to communication with the rest of the team and to mutual appreciation and support [41]. The midwifery model of care in pregnancy and low-risk childbirth, therefore, has advantages over medicalised models of care. These include a positive childbirth experience (PCE) for women, which in turn increases their satisfaction and that of their midwives [12,42].

### Limitations

This study has some limitations that are typical of qualitative research designs. Firstly, there is a risk of personal bias on the part of the researchers in the study. To avoid this, we used a triangulation of method and researcher. Secondly, there is the issue of participants’ own bias in their responses when discussing issues related to job satisfaction in their workplace. To overcome this, we conducted peer focus groups and prepared semi-structured questions that led to open-ended responses. In this way, participants could feel free to respond without being judged by the interviewers or their peers. Thirdly, convenience sampling was used, where participants were selected on the basis of availability and accessibility, which may result in an unrepresentative sample. To minimise this potential bias, we increased the sample size and selected participants from all three levels of hospital care. Although the study was mainly carried out with Spanish midwives, we conducted an in-depth interview with an Argentinian midwife in an effort to draw comparisons and minimise these differences. Furthermore, the in-depth analysis of the midwives’ comments and the broad socio-demographic profile of the participants allows for the transferability of the study.

Although the primary focus of this study was exploring midwives’ perspectives on the factors influencing their job satisfaction in delivery rooms, internal nurse residents were also included as participants. This decision is based on the fact that, while they are not midwives, internal nurse residents have direct and continuous exposure to clinical practice, working under the supervision of midwives. This variability in supervision may influence their perception of clinical practice and, consequently, their job satisfaction in obstetric settings. However, this inclusion may be seen as a limitation, as their perspectives may differ significantly from those of experienced midwives, potentially affecting the homogeneity of the sample. Nonetheless, their contributions were considered to enrich the analysis due to their unique role and fresher perspective on clinical practice.

## 5. Conclusions

This study highlights the key factors influencing midwives’ job satisfaction in the birthing room, including maternal satisfaction, obstetric intervention, professional skills, autonomy, team support, and working conditions. It emphasises the reciprocal relationship between women’s satisfaction with childbirth and midwives’ job satisfaction, underlining the need to focus on women’s experiences to enhance the quality of perinatal care.

We recommend that future research focus on the development of a tool that measures midwifery job satisfaction and takes into account the specialisms of the professionals involved in the process of labour and childbirth, with the topics identified in the testimonies collected. This instrument could include the following dimensions: appropriate development of competencies in the work area, support during the labour process, mutual satisfaction between the woman and the midwife, relationships with the multidisciplinary team, relationships with colleagues, prenatal education for the labour process, and the work-related factors influencing performance.

In this way, and based on the findings of this study, organisations involved in maternity care should consider implementing regular assessments of midwifery job satisfaction using the proposed tool. This would help identify areas where midwives feel supported and areas requiring improvement. Organisations should also invest in continuous professional development programmes to enhance the competencies of midwives, with a particular focus on improving autonomy and reducing unnecessary obstetric interventions. Strengthening team dynamics and improving the relationships between midwives and other healthcare professionals could further enhance job satisfaction and reduce burnout. Ultimately, this will ensure high-quality maternity care.

## Figures and Tables

**Figure 1 healthcare-13-01318-f001:**
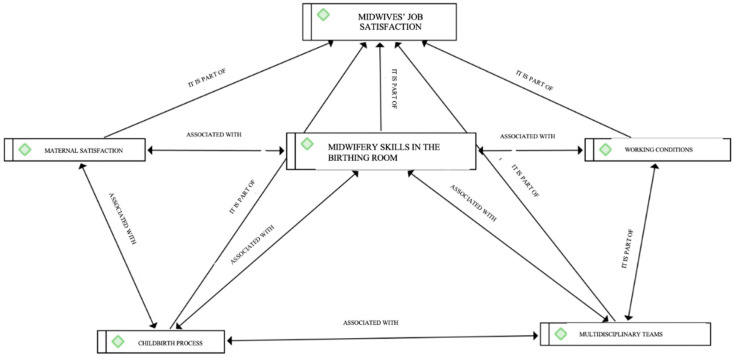
Interrelationships between the variables that contribute to midwives’ job satisfaction in birthing rooms.

**Table 1 healthcare-13-01318-t001:** Socio-demographic characteristics of the study participants.

Code (DG/IDI) *	Midwife/RIN	Age	Contract Type
FG1-P1	Midwife	49	Permanent
FG1-P2	Midwife	30	Temporary
FG1-P3	Midwife	44	Permanent
FG1-P4	Midwife	32	Temporary
FG1-P5	Midwife	40	Temporary
FG1-P6	Midwife	58	Permanent
FG2-P8	Midwife	47	Permanent
FG2-P9	Midwife	38	Temporary
FG2-P10	Midwife	50	Permanent
FG2-P11	Midwife	60	Permanent
FG3-P12	Midwife	66	Permanent
FG3-P13	Midwife	39	Permanent
FG3-P14	Midwife	34	Temporary
FG3-P15	Midwife	45	Permanent
FG4-P16	Midwifery RIN	24	Training
FG4-P17	Midwifery RIN	24	Training
FG4-P18	Midwifery RIN	24	Training
FG4-P19	Midwifery RIN	25	Training
FG4-P20	Midwifery RIN	35	Training
FG4-P21	Midwifery RIN	24	Training
FG4-P22	Midwifery RIN	24	Training
FG4-P23	Midwifery RIN	24	Training
FG4-P24	Midwifery RIN	24	Training
FG4-P25	Midwifery RIN	25	Training
IDI-P7	Midwife	48	Permanent

* FG/IDI: focus group/in-depth interview.

**Table 2 healthcare-13-01318-t002:** Measures that contribute to maternal and midwife satisfaction.

Measures that Contribute to Maternal and Midwife Satisfaction	Verbatim
**Holistic support**	*“In the whole person (…) supporting her everything she needs…”* (P20)
**Respect for women’s autonomy**	*“Being accompanied by the person of her choice the whole time…* (P7)
**Need for evidence-based learning**	*“(…) if you go to antenatal classes and you don’t realise that social networks can be full of misinformation (…)* (P24).
**Minimisation/influence of interventionism**	*“In the case of instrumental births, for example… (…) and then we see the interventionism of… well, some professionals… it can lead to lower satisfaction…”* (P23)
**Humanisation of surgical or interventionist procedures**	*“And, above all, that the woman’s satisfaction… that what the woman feels… whether it’s a ventouse birth, or a caesarean… is positive… (…)* (P13)

## Data Availability

Due to the ethical obligations of the authors, the data used in this study are not available. The authors are not authorised to disclose the data to anyone not involved in the project, which has been approved by the Ethics Committee.

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
