# Peer review of "Job Satisfaction Among Midwives in High-Intervention Birthing Rooms: A Qualitative Phenomenological Study"

_healthcare, 2025, doi:10.3390/healthcare13111318_

Round 1

Reviewer 1 Report

Comments and Suggestions for Authors

Dear Authors, thank you for your submission.

Your paper explores an important topic—the job satisfaction of midwives in birthing rooms, which has direct implications for maternity care quality and workforce sustainability. The use of hermeneutic phenomenology is appropriate for capturing the lived experiences of midwives, and your adherence to COREQ enhances the trustworthiness of the study.

Your study design is well-described, and your results are clearly presented and thematically structured. The interrelationship model presented in Figure 1 adds conceptual clarity, and the integration of verbatim quotes strengthens the analysis. The discussion is well-referenced and effectively connects your findings to existing literature.

However, I encourage you to consider the following minor revisions to improve clarity and impact:

1) Some redundancy exists between results and discussion. Consider synthesizing these sections more tightly to avoid repetition and enhance the analytic depth.

2) Language should be reviewed for grammatical precision and fluency. For example, in some places, phrasing such as “job well-being do not correspond to the reality” should be corrected to “job well-being does not reflect reality.”

3) The discussion could benefit from more critical reflection on the limitations of including RINs in the sample, especially given their trainee status and potentially different perspectives. Please, address this point accordingly.

4) While the conclusion mentions the need for a job satisfaction measurement tool, consider briefly outlining the dimensions that such a tool might include, based on your findings. This may be your future research, since this topic is rich with potential.

Your work is highly relevant and makes a valuable contribution to the literature on midwifery, workforce well-being, and obstetric care models. I support the publication of this paper pending minor revisions.

Author Response

Reviewer 1 Comments to the Author

Dear Reviewer:

We sincerely appreciate your constructive comments on our article. We have taken each of your points into consideration and have made the following modifications:

  • Comment 1. Some redundancy exists between results and discussion. Consider synthesizing these sections more tightly to avoid repetition and enhance the analytic depth.
    • Response: thank you very much for your comment regarding the manuscript. We have revised the discussion section as suggested, reducing the redundancies identified in relation to the results. Furthermore, we have restructured the section to enhance its clarity and coherence and have expanded upon certain analysis. These changes are evident throughout the discussion
  • Comment 2. Language should be reviewed for grammatical precision and fluency. For example, in some places, phrasing such as “job well-being do not correspond to the reality” should be corrected to “job well-being does not reflect reality.”
    • Response: thank you for your comment. We have reviewed the text to ensure grammatical precision and fluency, as requested.
  • Comment 3. The discussion could benefit from more critical reflection on the limitations of including RINs in the sample, especially given their trainee status and potentially different perspectives. Please, address this point accordingly.
    • Response: we greatly appreciate your comment and the opportunity to address this aspect. We acknowledge that the inclusion of RINs in the sample could have influenced the results, given that their trainee status may bring different perspectives compared to midwives with more professional experience. In this regard, we have added a more critical reflection in the discussion section, considering the potential implications of this heterogeneity in the experiences and perceptions of the participants. You can find this change in the Discussion section. We have also included a paragraph in the Discussion section highlighting the potential limitations of our manuscript.
  • 4. While the conclusion mentions the need for a job satisfaction measurement tool, consider briefly outlining the dimensions that such a tool might include, based on your findings. This may be your future research, since this topic is rich with potential.
    • Response: thank you for your comment. We fully agree on the importance of a job satisfaction measurement tool. We have added a brief comment in the conclusion section regarding the areas that could be included in the new measurement tool.

Reviewer 2 Report

Comments and Suggestions for Authors

1. “Job satisfaction in midwives working in birthing rooms. An exploratory study.” Please revise the title of the paper. For instance:

“Job satisfaction in midwives working in birthing rooms: An exploratory study”

2. Please rewrite the following sentence for better understanding.

“Therefore, it is possible that not all areas affecting their job satisfaction are being taking into account and this job well-being do not correspond to the reality.”

3. For any research paper, whether qualitative or quantitative, the research gap and novelty of the study are crucial. However, the introduction section lacks a clear explanation of the research gap and the study's novelty. Therefore, we request a clear explanation of the research gap and the study's novelty.

4. Please discuss the population size of the midwives and justify the sample size. Equally, state the reasons for using the convenience sampling technique in this study.

5. Please discuss the interview protocol in detail. How many questions are there? How were all the questions selected?

6. What does “EP-P7” mean? Please explain the coding of the interviewee. What is the difference between GD and EP? Please justify.

7. It would be excellent if the authors offered practical implications separately based on the study findings, which will guide the organizations.

Author Response

Reviewers 2's comments:

Dear Reviewer:

We sincerely appreciate your constructive comments on our article. We have taken each of your points into consideration and have made the following modifications:

  • Comment 1. “Job satisfaction in midwives working in birthing rooms. An exploratory study.” Please revise the title of the paper. For instance: “Job satisfaction in midwives working in birthing rooms: An exploratory study”
    • Response: we sincerely thank you for your comment. We have revised the title to more accurately reflect the qualitative and phenomenological-hermeneutic nature of the research, as well as the focus on midwives working in high-intervention birthing settings.
  • Comment 2. Please rewrite the following sentence for better understanding. “Therefore, it is possible that not all areas affecting their job satisfaction are being taking into account and this job well-being do not correspond to the reality.”
    • Response: we appreciate your feedback. The sentence you highlighted has been revised to enhance its clarity and ensure better understanding (lines 66-70).
  • Comment 3. For any research paper, whether qualitative or quantitative, the research gap and novelty of the study are crucial. However, the introduction section lacks a clear explanation of the research gap and the study's novelty. Therefore, we request a clear explanation of the research gap and the study's novelty.
    • Response: we are grateful for your comment. The text has been modified to provide a clearer explanation of the knowledge gap this study seek to fill (lines 71 to 76).
  • Comment 4. Please discuss the population size of the midwives and justify the sample size. Equally, state the reasons for using the convenience sampling technique in this study.
    • Response: Thank you for your comment. Regarding the population size and sampling strategy, we appreciate the opportunity to clarify and strengthen the methodology section. convenience sampling strategy was used due to the exploratory and qualitative nature of the study. This approach allowed us to recruit participants who were accessible and willing to share their experiences, which is appropriate for phenomenological research where depth and richness of data are prioritized over generalizability. The sample size of 25 participants was determined based on the principle of data saturation, as outlined by Malterud et al. (2016), and we ceased recruitment when no new themes were emerging from the data. This supports the appropriateness of our sample size in relation to the study design and objectives. We have added this clarification in the manuscript (see Methods section, subsection 2.2 Participants).

  • Comment 5. Please discuss the interview protocol in detail. How many questions are there? How were all the questions selected?
    • Response: Thank you for your comment. We appreciate the opportunity to clarify the interview protocol used in this study. As presented in Table 1, a semi-structured interview guide was developed, comprising nine thematic areas related to midwives' job satisfaction. Each theme included one or more guiding questions. The total number of initial guiding questions was 17; however, additional follow-up or probing questions were used depending on the flow of each focus group discussion. The interview guide was developed based on a review of relevant literature and existing theoretical frameworks on job satisfaction, professional autonomy, working conditions, and childbirth care. This theoretical grounding ensured that the questions addressed key domains known to influence midwives’ job satisfaction. The guide was reviewed by experts in qualitative research and midwifery to enhance its clarity and relevance. We have now included a more detailed description of the interview protocol in the manuscript (see section 2.3 Data Collection Procedure).

  • Comment 6. What does “EP-P7” mean? Please explain the coding of the interviewee. What is the difference between GD and EP? Please justify.
    • Response: thank you very much for your comment. The acronyms you referred to correspond to the Spanish terms for “focus groups” and in-depth interviews”. We have replaced them with their English equivalents and included a table footnote providing clarification.
  • Comment 7. It would be excellent if the authors offered practical implications separately based on the study findings, which will guide the organizations.
    • Response: thank you very much for your comment. We have included a few lines discussing the practical implications that our results could have for organizations.

Reviewer 3 Report

Comments and Suggestions for Authors

Dear Authors,

Firstly, I would like to sincerely thank you for your submission. Your work addresses a relevant and important topic — midwives’ job satisfaction in birthing rooms with moderate to high levels of obstetric intervention. It is evident that a great deal of effort and dedication have gone into this study.

However, after careful review, I must kindly highlight two major issues that must be addressed before the manuscript can be considered for further evaluation:

Manuscript Structure:

The current structure of the manuscript does not fully align with the standard template required by the journal (e.g., Healthcare or similar). Although the main sections (Introduction, Methods, Results, Discussion) are present, they are not adequately organised. Some sections are overly descriptive, while others mix methods, results, and discussion without clear separation. Furthermore, essential elements such as methodological rigour, analytical framework clarity, and limitations are not sufficiently emphasized.

English Language and Style:

The manuscript requires substantial revision to improve English usage.

Please understand that these recommendations are made with the sole purpose of helping you strengthen the quality of your article, ensuring that your important scientific contributions are conveyed with maximum clarity and academic rigour.

Once these changes are made, I am confident that your work will have greater impact and a higher chance of acceptance.

Detailed Section-by-Section Comments

Comment 1. You should consider modifying the title to clearly reflect the qualitative and phenomenological-hermeneutic nature of the research, as well as the focus on midwives in settings with high obstetric intervention. This will improve clarity and attract the appropriate readership.

Recommend

"Job Satisfaction among Midwives in High-Intervention Birthing Rooms: A Qualitative Phenomenological Study"

Comment 2 – Abstract

The abstract is generally well-structured but requires several revisions to meet scientific publication standards.

You should:

Correct the typographical error: “objetives” → “objectives”.

Add the philosophical and methodological foundation (e.g., “phenomenological-hermeneutic design based on Heideggerian philosophy”).

Clarify participant selection, specifying the inclusion of both midwives and resident intern nurses (RINs), and justify briefly.

Expand on the practical implications of your findings in the conclusion sentence of the abstract.

Recommend ending sentence for the abstract:

“These findings may inform healthcare management strategies to reduce burnout and improve working conditions in maternity care settings.”

Comment 3 – Introduction

The introduction sets the context, but is repetitive in several places and lacks a critical framing of the problem.

You should:

  • Avoid clustering too many references without individual discussion (e.g., “[3,4]” or “[24–28]”).
  • Clearly define the gap in literature that your study aims to address.
  • Better justify the choice of a qualitative hermeneutic approach at the end of the introduction.
  • End with a concise and clear research aim or guiding question.

Comment 4 – Materials and Methods

This section includes valuable information but lacks clarity and completeness in several key areas.

You should:

4.1 Study Design

Clarify how Heideggerian phenomenology guided the analysis process.

Explain how the depth of lived experiences was achieved in focus groups lasting only 15–25 minutes (which is short for phenomenological studies).

4.2 Participants

Justify the inclusion of RINs (trainees) in the same sample as practicing midwives, especially given their different levels of responsibility.

Clarify the rationale for including one participant from Argentina (potential for cultural/contextual bias).

Fix the grammatical error: “the do not require…” → “they do not require…”

4.3 Data Collection

The interview guide (Table 1) is too long and detailed for inclusion in the main manuscript. It should be moved to a supplementary file or appendix.

Provide details on the role of the moderator, her relationship with participants, and how bias was mitigated.

State clearly how data saturation was confirmed.

4.4 Data Analysis

You should add detail about the number of coders and whether inter-rater reliability or consensus discussions were conducted.

Explain how categories were developed from codes and refined over time.

4.5 Rigour

The triangulation process is mentioned, but needs elaboration: What roles did each researcher play? How were discrepancies resolved?

Clarify the credibility, transferability, and confirmability strategies used.

4.6 Ethics

Provide the full ethics approval reference number.

Indicate adherence to the Declaration of Helsinki or equivalent guidelines.

Comment 5 – Results

The results are rich and well-illustrated with verbatim quotes, but the section is too long and includes repetition.

You should:

Avoid repetition of ideas across categories (e.g., “maternal and midwife satisfaction” is repeated excessively).

Ensure that Table 2 is complete (e.g., correct the missing header in the “Contract Type” column).

Limit the number of quotations—only include those that add analytical value.

Consider reorganising subheadings for clarity and flow.

Comment 6 – Discussion

The discussion is insightful but lacks structure and critical reflection in some areas.

You should:

Clearly separate summary of findings from interpretation and implications.

Ensure all references are correctly numbered and included in the reference list (e.g., citation “(46)” appears without a corresponding entry).

The term “maternal and midwife satisfaction” needs to be formally defined or supported with literature.

Discuss the methodological limitations more explicitly (e.g., short duration of focus groups, inclusion of RINs).

Deepen the reflection on power dynamics and obstetric violence using existing literature.

Comment 7 – Conclusions

The conclusions restate findings but lack critical synthesis and actionable recommendations.

You should:

Summarize the main contribution of the study in 1–2 strong sentences.

Offer concrete implications for practice, policy, and future research.

Expand the idea of developing a measurement tool for midwifery job satisfaction: who should develop it? What dimensions should it include

Best Regards

Comments on the Quality of English Language

The quality of the English language needs major revision.
Numerous grammatical errors, unclear sentences, and structural weaknesses make it difficult to fully appreciate the value of the study. The manuscript requires professional English language editing.

Author Response

Reviewers 3's comments:

Reviewer:

We sincerely appreciate your constructive comments on our article. We have taken each of your points into consideration and have made the following modifications:

  • Comment 0.1. Manuscript Structure: The current structure of the manuscript does not fully align with the standard template required by the journal (e.g., Healthcare or similar). Although the main sections (Introduction, Methods, Results, Discussion) are present, they are not adequately organised. Some sections are overly descriptive, while others mix methods, results, and discussion without clear separation. Furthermore, essential elements such as methodological rigour, analytical framework clarity, and limitations are not sufficiently emphasized.
    • Response: we appreciate your detailed feedback regarding the structure of the manuscript. We have thoroughly revised the manuscript to align with the journal structure and modified the structure of the main sections to align more closely with the specific characteristics of each.
  • Comment 0.2. English Language and Style: The manuscript requires substantial revision to improve English usage.
    • Response: thank you very much for your comment. We have reviewed the manuscript regarding language and English style. We sincerely hope that the quality has been improved.
  • Comment 1. You should consider modifying the title to clearly reflect the qualitative and phenomenological-hermeneutic nature of the research, as well as the focus on midwives in settings with high obstetric intervention. This will improve clarity and attract the appropriate readership. Recommend "Job Satisfaction among Midwives in High-Intervention Birthing Rooms: A Qualitative Phenomenological Study"
    • Response: we sincerely thank you for your insightful comment and valuable suggestion regarding the title of the manuscript. We have revised the title to more accurately reflect the qualitative and phenomenological-hermeneutic nature of the research, as well as the focus on midwives working in high-intervention birthing settings.
  • Comment 2 – Abstract. The abstract is generally well-structured but requires several revisions to meet scientific publication standards. You should: Correct the typographical error: “objetives” → “objectives”. Add the philosophical and methodological foundation (e.g., “phenomenological-hermeneutic design based on Heideggerian philosophy”). Clarify participant selection, specifying the inclusion of both midwives and resident intern nurses (RINs), and justify briefly. Expand on the practical implications of your findings in the conclusion sentence of the abstract. Recommend ending sentence for the abstract: “These findings may inform healthcare management strategies to reduce burnout and improve working conditions in maternity care settings.
    • Response: we appreciate your feedback and suggestion you have provided. We have made the suggested changes in the abstract These are highlighted in yellow.
  • Comment 3 – Introduction. The introduction sets the context but is repetitive in several places and lacks a critical framing of the problem. You should: Avoid clustering too many references without individual discussion (e.g., “[3,4]” or “[24–28]”). Clearly define the gap in literature that your study aims to address. Better justify the choice of a qualitative hermeneutic approach at the end of the introduction. End with a concise and clear research aim or guiding question.
    • Response: thank you very much for your comment. We have implemented the suggested changes in the manuscript: we have avoided clustering citations unnecessarily, modified the text to avoid unnecessary repetition, added a justification at the end of the section to explain the choice of a qualitative hermeneutic approach, revised the research aim to make it more concise and clear and defined the gap in literature that our study aims to address. All changes have been highlighted in yellow.
  • Comment 4 – Materials and Methods. This section includes valuable information but lacks clarity and completeness in several key areas.
    • Comment 4.1. Study Design. Clarify how Heideggerian phenomenology guided the analysis process. Explain how the depth of lived experiences was achieved in focus groups lasting only 15–25 minutes (which is short for phenomenological studies).
      • Response: Thank you for your thoughtful comment. We acknowledge that phenomenological research, particularly within a Heideggerian framework, typically seeks to access the depth and complexity of lived experience. In our study, Heideggerian phenomenology served as the guiding philosophical and methodological orientation by focusing on how midwives interpret their role and existence in the birthing room. Rather than aiming for objective descriptions, we sought to interpret how participants construct meaning in relation to their professional identity, autonomy, and context of care — that is, their Being-in-the-world as midwives. Regarding the duration of focus groups (15–25 minutes), we are aware that this may appear short for phenomenological exploration. However, these sessions were conducted within the birthing room setting, and their timing was determined by participants’ real-time availability during work shifts. The nature of the labour ward environment required flexible, adaptive scheduling, where group discussions were embedded within the rhythm of clinical practice. Despite the limited duration, the discussions were intense, focused, and structured around a theoretically informed interview guide, which facilitated the emergence of rich narratives. Additionally, we used field notes, researcher reflexivity, and one in-depth interview to complement and deepen our understanding of the midwives’ lived experiences.We have clarified this rationale in the manuscript (see sections 2.1 Study Design and 2.3 Data Collection Procedure).
    • Comment 2 Participants. Justify the inclusion of RINs (trainees) in the same sample as practicing midwives, especially given their different levels of responsibility. Clarify the rationale for including one participant from Argentina (potential for cultural/contextual bias). Fix the grammatical error: “the do not require…” → “they do not require…”
      • Response: Regarding the inclusion of midwifery trainees (Resident Intern Nurses, RINs), we acknowledge that they are at a different stage of professional development compared to fully qualified midwives. However, in the Spanish healthcare system, second-year RINs are integrated into clinical practice with a high degree of autonomy and responsibility, especially in birthing rooms. Their roles are similar to those of licensed midwives and they participate in the same routines, care decisions, and team dynamics. Including them allowed us to capture the perspective of professionals who are already immersed in the labour ward setting and who, despite being in training, experience the same working conditions and satisfaction/dissatisfaction factors. For transparency, we have also disaggregated the data in the results section and clarified their contribution to the focus groups.
      • Response: As for the inclusion of a midwife from Argentina, this was a strategic choice to enrich the analysis by adding a broader perspective. While the study is focused on the Region of Murcia (Spain), this single in-depth interview was intended to introduce a contrasting viewpoint that might highlight context-specific aspects of satisfaction or dissatisfaction. We recognise the possibility of contextual variation, and for this reason, the data from the Argentinian participant were used only as a complementary, illustrative example and were not weighted equally with the focus group findings. We have clarified this in the manuscript to avoid any misunderstanding about its role in the analysis.
    • Comment 3 Data Collection. The interview guide (Table 1) is too long and detailed for inclusion in the main manuscript. It should be moved to a supplementary file or appendix. Provide details on the role of the moderator, her relationship with participants, and how bias was mitigated. State clearly how data saturation was confirmed.
      • Response: We appreciate the reviewer’s constructive feedback regarding the data collection section of our manuscript. Below, we address the points raised and indicate the corresponding changes to be made:
        • Interview Guide (Table 1): As suggested, we will move the detailed interview guide (Table 1) to appendix.
        • In response to your comment, we have included detailed information in the Methodology section about the role of the moderator, her relationship with the participants, and the strategies used to mitigate potential bias. Additionally, we have clearly clarified how data saturation was confirmed during the data collection and analysis process. These changes aim to improve understanding and transparency regarding these methodological aspects.
      • Comment 4.4 Data Analysis. You should add detail about the number of coders and whether inter-rater reliability or consensus discussions were conducted. Explain how categories were developed from codes and refined over time.
        • Response: Thank you for your comment. We have revised the Data Analysis section to include further detail regarding the coding and categorization process. Specifically:
          • Number of coders and reliability process: Two researchers independently coded the transcripts using a thematic analysis approach. Following the initial coding, they met to compare results and discuss discrepancies. Any disagreements were resolved through discussion and, when necessary, by consulting a third member of the research team. Decisions were guided by theoretical sensitivity and supported by relevant literature to ensure coherence and analytical rigor.
          • Development and refinement of categories: Codes were initially generated inductively from the data and then grouped into preliminary categories based on thematic similarity. Through iterative discussions and constant comparison across transcripts, these categories were further refined, merged, or reorganized to better reflect the underlying patterns in participants’ narratives. This process contributed to the development of final themes that are grounded in the data and aligned with the study’s aims.
        • Comment 5 Rigour. The triangulation process is mentioned, but needs elaboration: What roles did each researcher play? How were discrepancies resolved? Clarify the credibility, transferability, and confirmability strategies used.
          • Response: Thank you for your valuable feedback. In response, we have expanded the description of the triangulation process and the strategies used to ensure the study's trustworthiness The changes are applied in the analysis and rigor sections. Specifically:
            • Roles of the researchers: The triangulation involved multiple researchers with complementary profiles. The first and second authors independently conducted the initial coding and thematic analysis. The third author contributed as an external reviewer, examining the coded data and providing critical feedback to refine theme development. This process ensured that different perspectives enriched the interpretation and minimized the influence of individual bias.
            • Resolution of discrepancies: Coding discrepancies between the two primary analysts were discussed in joint sessions. When disagreements could not be resolved through dialogue, a third researcher was consulted. Decisions were made based on theoretical justification and supported by relevant literature. This collaborative process contributed to analytic rigor and internal coherence.
            • Credibility, transferability, and confirmability: Credibility was ensured through investigator triangulation, prolonged engagement with the data, and peer debriefing sessions throughout the analytic process. Transferability was supported by providing rich, contextualized descriptions of participants and settings, allowing readers to assess the applicability of findings to other contexts. Confirmability was enhanced by maintaining an audit trail of analytical decisions, memos, and code development, as well as by practicing reflexivity throughout the study to recognize and bracket potential researcher biases.

  • Comment 4.6 Ethics, Provide the full ethics approval reference number. Indicate adherence to the Declaration of Helsinki or equivalent guidelines.
    • Response: Thank you for your comment. We confirm that the study was approved by the Ethics Committee of the University of Murcia, and the full ethics approval reference number is CI 4065, as already stated in the manuscript. This is the official coding format used by the Ethics Committee of our institution. Additionally, we confirm that the study was conducted in accordance with the ethical principles outlined in the Declaration of Helsinki, including informed consent, voluntary participation, and confidentiality of data. We have now explicitly added this reference in the Ethical Considerations section to clarify compliance with international ethical standards.
  • Comment 5 – Results. The results are rich and well-illustrated with verbatim quotes, but the section is too long and includes repetition. You should: Avoid repetition of ideas across categories (e.g., “maternal and midwife satisfaction” is repeated excessively). Ensure that Table 2 is complete (e.g., correct the missing header in the “Contract Type” column). Limit the number of quotations—only include those that add analytical value. Consider reorganising subheadings for clarity and flow.
    • Response: thank you very much for your comment. We have removed the repeated ideas found throughout the results We have corrected Table 2 as you suggested, limited the number of verbatim quotes to those essential, and reorganised some headings to improve the clarity and flow of the text.
  • Comment 6 – Discussion The discussion is insightful but lacks structure and critical reflection in some areas. You should: Clearly separate summary of findings from interpretation and implications. Ensure all references are correctly numbered and included in the reference list (e.g., citation “(46)” appears without a corresponding entry). The term “maternal and midwife satisfaction” needs to be formally defined or supported with literature. Discuss the methodological limitations more explicitly (e.g., short duration of focus groups, inclusion of RINs). Deepen the reflection on power dynamics and obstetric violence using existing literature.
    • Response: thank you very much for your comment. We have made the necessary revisions to the manuscript. We have separated the summary of the findings from the interpretations and implications, reviewed the reference list and corrected the issue with reference 46, defined the term maternal and midwife satisfaction based on our findings and the existing literature. We have also included methodological limitations and provided a more in-depth reflection on the issues raised. All changes can be found throughout de discussion
  • Comment 7 – Conclusions. The conclusions restate findings but lack critical synthesis and actionable recommendations. You should: Summarize the main contribution of the study in 1–2 strong sentences. Offer concrete implications for practice, policy, and future research. Expand the idea of developing a measurement tool for midwifery job satisfaction: who should develop it? What dimensions should it include.
    • Response: we thank you for your valuable suggestions. We have revised and restructured the conclusions section to provide a more critical synthesis of the findings and to offer more concrete practical recommendations. In the revised version, we have summarised the main contribution of the study in clear sentences and detailed the implications for practice and future research. Additionally, we have expanded on the idea of developing a measurement tool for midwifery job satisfaction, specifying the dimensions it should include. We believe these changes enhance the clarity and applicability of the study (conclusion section).

Round 2

Reviewer 3 Report

Comments and Suggestions for Authors

Dear Authors,

Thank you for your comprehensive revisions. The manuscript now shows clear improvements in structure, clarity, and methodological transparency. Most reviewer concerns have been addressed, and the text meets the journal’s standards.

Best Regards